



# Feasibility study for 100% renewable energy microgrids in Switzerland

Sarah Barber[1], Simon Boller[1], and Henrik Nordborg[1]

[1]University of Applied Sciences Rapperswil, Oberseestrasse 10, 8640 Rapperswil, Switzerland

**Correspondence:** Sarah Barber (sarah.barber@hsr.ch)

**Abstract.** The growing worldwide level of renewable power generation requires innovative solutions to maintain grid reliability and stability, due to their variability and uncertainty. As well as stabilising the grid, renewable microgrids are also attractive solutions for regions wishing to produce green electricity independently from the grid, saving potentially high cable laying and grid connection costs. Switzerland does not yet have a large number of installed wind turbines, and despite the ambitious Energy Strategy 2050, not a single wind turbine was installed in 2018. This lack of progress is mainly due to large delays, costs and risks associated with the permitting procedure for wind turbines. The implementation of medium-sized wind turbines smaller than about 30 m may be a possible solution to this lack of progress, as they could experience an easier permitting procedure and higher acceptance among local residents. However, medium-sized wind turbines are less economically viable than larger wind turbines, and one method of making them more economically viable could be to combine them with photovoltaics (PV) and electricity storage into a renewable microgrid system. In this work, twelve sites in Switzerland are chosen for a 100% renewable energy microgrid feasibility study. For all of these sites, a combination of wind and PV performs consistently better than wind only and PV only. This is due to the fact that wind speeds are often higher when the solar radiation is low, and vice versa. The combination of wind and PV ensures a more constant coverage of renewable energy production and therefore is more efficient. Five of the sites are found to be potentially economically viable, if investors would be prepared to make extra investments between 0.05 $/kWh and 0.2 $/kWh or between $5 million and $20 million upfront for green electricity independence. The Self-Sufficiency Ratio (SSR) is found to range between 1 and 2 for each site, reflecting the extra installed capacity required in order to fully cover every hour of demand in island operation. This could be decreased by connecting to the grid at times of low wind and solar resource and high demand. For the *Wind and PV* combination, halving battery capital costs reduces Costs of Electricity (COE) by 11%, decreases the number of wind turbines by 1% and reduces SSR by 1%. Halving the wind turbine capital costs reduces COE by 30%, increases the number of wind turbines by 16% and increases the SSR by 16%. Reducing the PV capital costs by 50% reduces COE by 8%, decreases the number of wind turbines by 39% and decreases the SSR by 19%. The actual implementation of 100% renewable energy microgrids with medium-sized wind turbines is found to be strongly limited by the area required by the wind turbines as well as by the total number of wind turbines that can be realistically implemented. Additionally, a case study for an extension to a High Performance Computing centre in Canton Glarus shows that a feasible solution is available that meets the requirements. Specific projects are being further examined on a case-to-case basis.





# 1 Introduction

The growing worldwide levels of renewable power generation requires innovative solutions to maintain grid reliability and stability, due to their variability and uncertainty. The implementation of microgrids in small regions can help maintain grid reliability and stability by intelligently storing or releasing electricity to the grid, depending on the loads and the grid require-
ments at any specific time. The United States Department of Energy Microgrid Exchange Group defines a microgrid as (Smith, 2012):

   *"A group of interconnected loads and distributed energy resources within clearly defined electrical boundaries that acts as a single controllable entity with respect to the grid. A microgrid can connect and disconnect from the grid to enable it to operate in both connected or island-mode".*

As well as stabilising the grid, renewable microgrids are also attractive solutions for regions wishing to produce green electricity independently from the grid, saving potentially high cable laying and grid connection costs. Existing renewable energy microgrids include for example Kodiak Island (ABB, 2019) and the Ohio Trucking Terminal (Burger, 2018). Switzerland does not yet have a large number of installed wind turbines (0.2% penetration level in 2016) and despite the ambitious Energy Strategy 2050, not a single wind turbine was installed in 2018 (SuisseEole, 2019). This lack of progress is mainly due to
large delays, costs and risks associated with the permitting procedure for wind turbines. The implementation of medium-sized wind turbines smaller than about 30 m may be a possible solution to this lack of progress, as they could experience an easier permitting procedure and higher acceptance among local residents. However, medium-sized wind turbines are less economically viable than larger wind turbines, kilowatt-scale wind turbines generally having average installed costs of about 4,400 $/kW compared to 1,400 $/kW for megawatt-size wind turbines (IRENA, 2012). One method of making the implementation
of medium-sized wind turbines more economically viable could be to combine them with photovoltaics (PV) and electricity storage into a renewable microgrid system. This can reduce the electricity costs due to the complementary nature of the wind and solar resource (solar radiation is often high when the wind speed is low, and vice versa) and at the same time be easier to implement in Switzerland due to increased acceptance or easier permitting procedures.

   To the knowledge of the authors, no study on the implementation of a renewable microgrid system in Switzerland has been
published. In this work, a feasibility study for several sites in Switzerland was carried out for a range of possible load profiles, and the results evaluated in terms of Cost Of Electricity, available area and Self-Sufficiency Ratio. The method is described in Section 2 and the results are shown and discussed in Section 3. The conclusions are presented in Section 4.

# 2 Method

This study was carried out using the commercial software *HOMER Pro* (Lilienthal, 2006). As shown in Fig. 1, it involved (1)
choosing possible sites based on wind speed, terrain and nearby villages, (2) obtaining and entering solar radiation and wind speed distribution data for each site, (3) defining the microgrid layout, set-up and components, (4) choosing a range of suitable





load profiles, (5) choosing the most suitable algorithm and (6) running the optimisation algorithm to calculate the optimal combination of PV, wind and storage. These steps are described in more detail below.

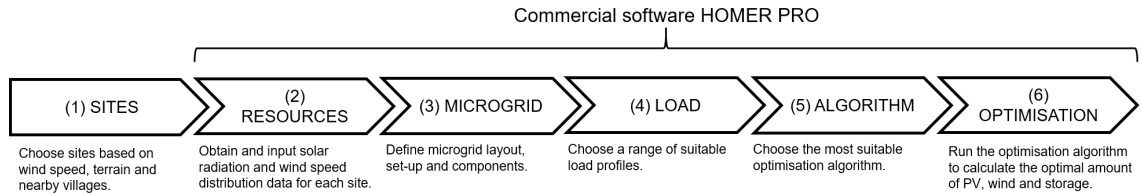

**Figure 1.** Method used for the feasibility study in this work.

## 2.1 Choice of sites

The sites studied in this work are shown in Fig. 2 and consisted of four locations in Canton St. Gallen (Grabs, Hemberg, Krinau,
Wattwil), eight other locations in the rest of Switzerland (Bad Ragaz, Braunwald, Feldis Neuiden, Les Pleiades, Peist, Stoos, Sumiswald, Zermatt) as well as two case study locations for a High Performance Computing centre located near Schwanden, Canton Glarus. The sites were chosen by considering their suitability for a renewable microgrid solution, by assessing parameters such as their location, their distance to communities or industrial areas, as well as the average wind speed at the site.

## 2.2 Input resources

For the **wind energy resource**, average yearly wind speeds were obtained at a height of 50 m above ground (the lowest available height) at each location from the Swiss Wind Atlas (*www.wind-data.ch*), which has a spatial resolution of 0.5 km. These values were then extrapolated down to hub height using the power law with an exponent of 0.14. Monthly variations were obtained at each site location from the *HOMER* database, which is based on data from 239 weather stations in the U.S. National Solar
Radiation Data Base (called the TMY2 database (NREL, 2019)). These values were then scaled linearly in order to achieve an annual average wind speed equal to the extrapolated hub-height wind speed obtained from the Swiss Wind Atlas.

Synthetic hourly wind data was generated by *HOMER Pro* using the Weibull k-factor, the one-hour autocorrelation factor, the diurnal pattern strength and the hour of the peak wind speed. The Weibull k-factor was estimated from the average wind speed, $V_0$, using the in-built linear assumption from the TMY2 data set (NREL, 2019) shown in equation 1.

$$k = 0.27 \times V_0 + 0.78 \tag{1}$$

The one-hour autocorrelation factor reflects how strongly the wind speed in one time step depends on the wind speed in previous time steps. Autocorrelation factors tend to be lower (0.70 - 0.80) in areas of complex topography and higher (0.90 - 0.97) in areas of more uniform topography. The value of the one-hour autocorrelation factor was estimated in *HOMER Pro*



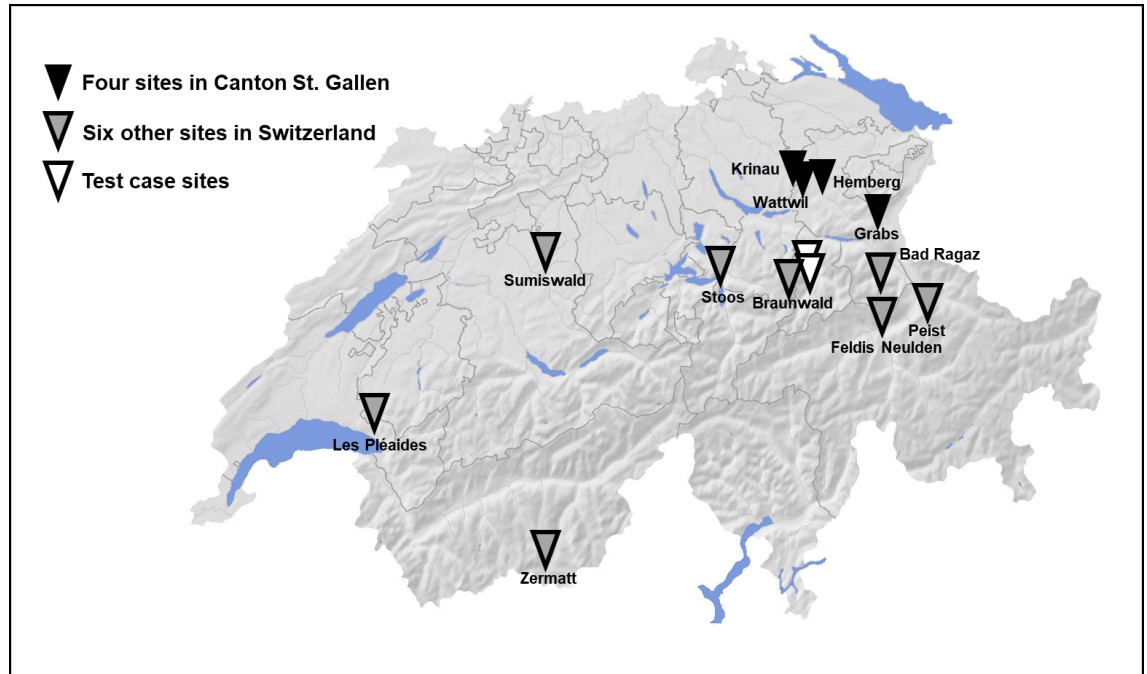

**Figure 2.** Sites studied for the feasibility study in this work (*www.wind-data.ch*).

based on the factors taken from the TMY2 data following a factoring-out of the diurnal data. The diurnal pattern strength is a number between 0 and 1 that reflects how strongly the wind speed tends to depend on the time of day. In this work, a cosinusoidal diurnal pattern was assumed, with the diurnal pattern strength defined as the ratio of the amplitude to the mean. The hour of the peak wind speed was chosen to correspond to the nearest measurement point from the TMY2 data.

5   For the **solar energy resource**, monthly values of the Solar Global Horizontal Irradiation (GHI) were downloaded from the NASA Surface Meteorology and Solar Energy Database based on the latitude and longitude, which averages these values between 1983 and 2005. GHI is the total solar radiation incident on a horizontal surface calculated by summing the Direct Normal Irradiance, the Diffuse Horizontal Irradiance, and ground-reflected radiation. A set of 8,760 hourly solar radiation values was then generated and synthesised using the Graham algorithm (Hollands, 1990), giving a realistic day-to-day and
10  hour-to-hour variability and auto-correlation.

## 2.3   Microgrid layout

The microgrid studied in this work consisted of a wind turbine, a generic flat plate PV, an idealised lithium-ion battery and a converter as shown in Fig. 3, with the properties shown in Table 1. The power curve of the Eocycle wind turbine was provided by *HOMER Pro*, which is required for the calculation of the electricity production later (Section 2.5).



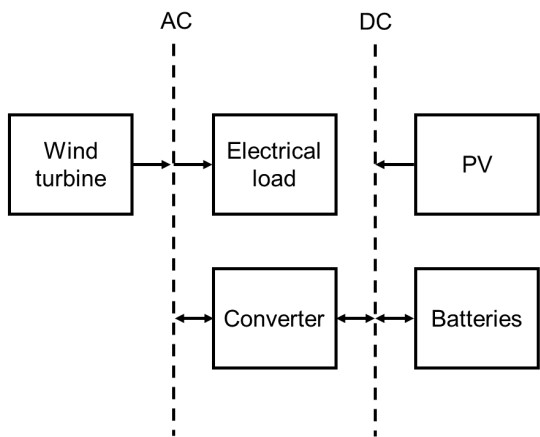

**Figure 3.** Schematic set-up of the microgrid.

**Table 1.** Properties of the microgrid components

| Component | Wind turbine | PV | Battery |
|---|---|---|---|
| **Type** | EO10 Eocycle | Generic flat PV | Standard Lithium-Ion |
| **Rated capacity** | 10 kW | 1 kW | 100 kWh |
| **Rotor diameter** | 15.81 m | - | - |
| **Hub height** | 16 m | - | - |
| **Capital costs ($)** | 4,400/kW (IRENA, 2012) | 3,000/kW (Gloor, 2018) | 200/kWh (Curry, 2017) |
| **Operating costs ($/year)** | 4,000 | 10 | - |

## 2.4  Load profiles

For the load profiles, a synthetic baseline load consisting of hourly values for each month was first created from a generic daily and seasonal profile as shown in Fig. 4. Next, seven different load profiles were created by scaling this base load based on assumptions regarding the total electricity demand per day as shown in Table 2. Larger daily loads were not examined due to the space taken up by the wind turbines and PV systems, as discussed further in Section 3.

## 2.5  Algorithm

The yearly electricity production for one wind turbine at each site was calculated by estimating the Weibull factors from the TMY2 data as described above and multiplying the resulting hub-height Weibull distribution with the power curve provided by the wind turbine manufacturer and 8,760 hours. The power curve was firstly corrected for density at the site according to



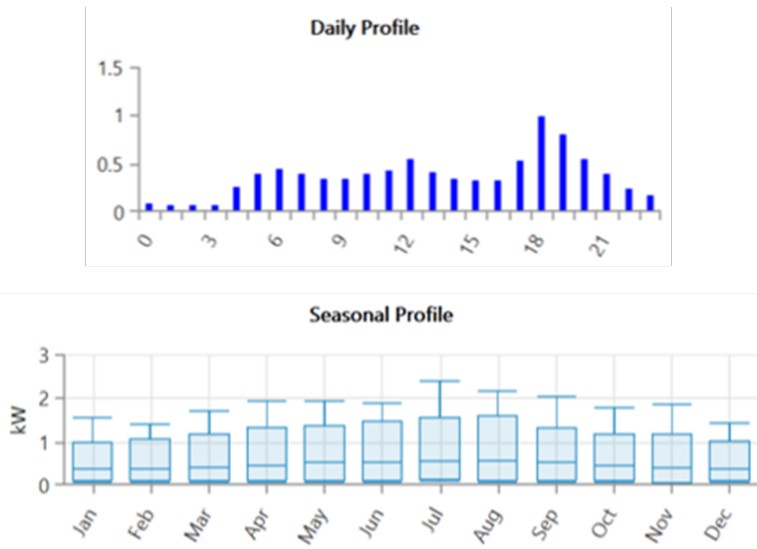

**Figure 4.** Generic load profiles used for the feasibility study.

**Table 2.** Different load profile sizes applied.

| Load profile | Electricity demand (kWh/day) | Equivalent number of family houses (ref) |
|---|---|---|
| 1 | 500 | 73 |
| 2 | 1,000 | 146 |
| 3 | 3,000 | 438 |
| 4 | 4,000 | 584 |
| 5 | 5,000 | 730 |
| 6 | 10,000 | 1,460 |
| 7 | 16,000 | 2,336 |

the International Electrotechnical Committee standard IEC 61400-12 (IEC). 3% availability losses, 5% wake effect losses and 2% electrical losses were deducted from the total.

Maximum Power Point Tracking (MPPT) was chosen for the PV system. Its yearly electricity production was obtained by summing power calculated in equation 2 for each one-hour period and multiplying this by 8,760 hours in one year.

5 $P_P = Y_P \times f_P \times (\bar{G}_T / \bar{G}_S)$ (2)





where $P_P$ = power output of the array (kW), $Y_P$ = rated capacity of the array (kW), $f_P$ = power de-rating factor (%), $\bar{G}_T$ = solar radiation incident on the PV array (kW/m$^2$), and $\bar{G}_S$ = incident radiation at standard test conditions (1 kW/m$^2$).

For each hour, the maximum amount of power that the storage bank could absorb as well as the maximum amount of power that it could discharge were calculated in order to decide if surplus power could be absorbed or provided, depending on the

demand and the renewable energy production. The maximum charge power varies from one time step to the next according to its state of charge and its recent charge and discharge history. Three separate limitations were placed on the storage bank's maximum charge power: the kinetic storage model (McGowan, 1993), the maximum charge rate and the maximum storage current.

For a nominal discount rate of 6% and a project lifetime of 20 years, the *HOMER Pro* software was used to choose the

optimum combination of number wind turbines, PV panels and batteries that minimises the Net Present Cost (NPC) of the project. Note that US Dollars ($) and Swiss Francs (CHF) are treated as interchangeable in this work, as the exchange rate is equal to about 0.99.

## 3   Results

### 3.1   Cost of Electricity (COE)

The optimum combination of wind turbines, PV panels and battery storage was calculated for each site for all seven load profiles defined in Section 2. This combination is referred to as *Wind and PV* in the remainder of the paper. For comparison purposes, the optimal combination of wind turbines and batteries (without PV, named *Wind only* here) and of PV and batteries (without wind turbines, named *PV only* here) were also calculated.

The resulting Cost of Electricity (COE) for load profile number 7 (the highest load, 16,000 kWh/day) for each site for these

three combinations is shown in Fig. 5. The current feed-in tariff for wind energy in Switzerland (0.2 CHF/kWh) (BFE, 2019) as well as the mean electricity price in Switzerland in 2018 (0.24 CHF/kWh) (Elcom, 2019) are shown as dashed and solid lines, respectively. It can be seen that the *Wind and PV* combination performs consistently better than *Wind only* and *PV only*. This is due to the fact that wind speeds are often higher when the solar radiation is low, and vice versa. The combination of wind and PV ensures a more constant coverage of renewable energy production and therefore is more efficient.

The results for the combined solution are shown in terms of extra electricity costs on top of the average electricity price in Fig. 6. The second y-axis shows the results converted to the extra investment that would be required to cover these increased costs if they were to be paid up-front. The five most economically viable sites are circled on the plot, requiring an extra investment ranging from 0.05 $/kWh to 0.2 $/kWh or $5 million to $20 million. The feasibility of these solutions and the willingness of investors to cover these extra costs are extremely project-specific and can depend on other factors such as extra

costs associated with traditional PV or large wind solutions, including new cable-laying or grid connections. Specific projects should be handled on a case-to-case basis in further work. The COE values could be further reduced by reducing the wind turbine, PV and battery costs.



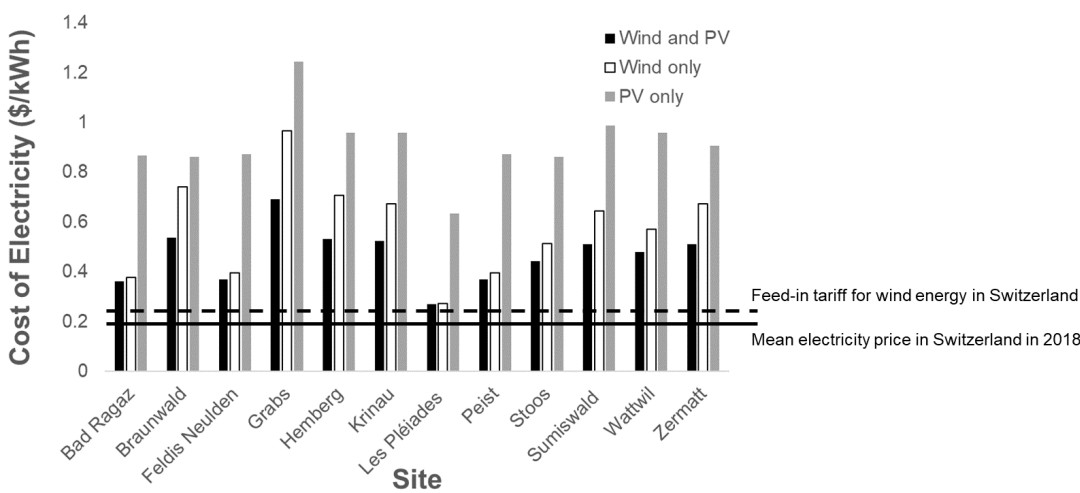

**Figure 5.** Cost of Electricity for all three combinations for all the sites for a load of 16,000 kWh/day.

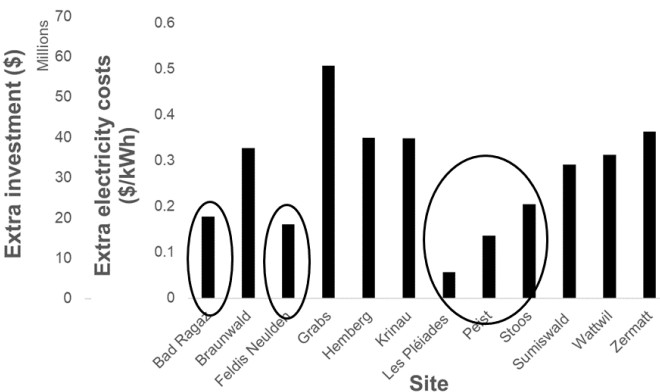

**Figure 6.** Extra electricity costs for combined wind turbines and PV for all the sites for a load of 16,000 kWh/day.

## 3.2 Self-Sufficiency Ratio (SSR)

The Self-Sufficiency Ratio (SSR) is defined as the ratio of actual electricity production to the demanded electricity production, as shown in equation 3.

$$SSR = E_A / E_D \qquad (3)$$





where $E_A$ = Actual wind and PV production (kWh) and $E_D$ = Production demanded (kWh).

In an ideal system, this value is equal to one all the time, and the electricity produced is always equal to the electricity demanded. If the actual production does not reach the demand, the value of SSR is below one. If the actual production is higher than the demand, the value of SSR is above one. For an island mode system, it is therefore important that SSR is above one at

all times. For a renewable-only island mode system, this inevitably leads to an average yearly SSR that is significantly higher than one, as the entire system has to be capable of covering the demand even at times where the wind and solar resources are low. The battery can cover these low-resource periods to some extent, but high battery costs can limit the total battery size at the expense of an over-supply of wind and PV production.

The calculated average yearly SSR for each site for the 16,000 kW/h load case is shown in Fig. 7. This shows, as expected,

that the SSR is significantly above one for all of the sites. For nearly all the sites, the *Wind and PV* combination has a lower SSR than *Wind only* and *PV only*. This is because the combination of wind and PV allows the set-up to more efficiently cover the production demands due to the complementary nature of the solar and wind resources. Furthermore, for nearly all the sites, the *Wind only* set-up has the highest SSR. This is because wind power fluctuates more over the year that solar power does. The value of SSR could be decreased for the cases studied here by connecting to the grid at times of low wind and solar resource

and high demand. Additionally, reducing battery costs are expected to have a large impact on the SSR reduction.

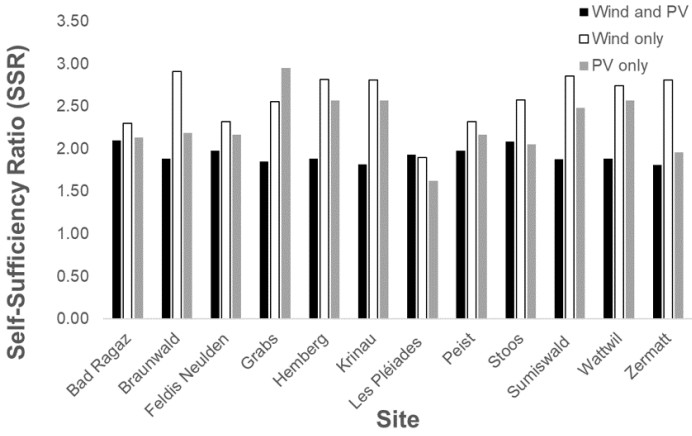

**Figure 7.** Self-sufficiency ratio for all the sites for a load of 16,000 kWh/day.

## 3.3 Sensitivity studies

In this section, the effects of reducing the battery costs, the capital cost of the wind turbines and the capital cost of the PV panels are investigated by examining the highest load case at the most favourable site, Les Pleiades. Each variable was altered separately.

As can be seen in Fig. 8(a), reducing the battery capital costs from 200 \$/kWh to 100 \$/kWh leads to only a small reduction in COE for all the *Wind only*, *PV only* and *Wind and PV* combinations. Halving the battery capital costs reduces COE by 11% for



the *Wind and PV* combination. Figure 8(b) shows that reducing the battery capital costs leads to a small decrease in the number of wind turbines chosen for the optimal solution, as expected, because more batteries are used if they are cheaper. Halving the battery capital costs decreases the number of wind turbines by only 1% for the *Wind and PV* combination. Additionally, Fig. 8(c) shows that reducing the battery capital costs leads to a decrease in SSR, as expected. This is because more battery storage

and fewer wind turbines lead to a more even matching of supply and demand. However, the decrease in SSR is much smaller than expected, especially for the *Wind only* and *Wind and PV* combinations - for the *Wind and PV* combination, halving the battery capital costs only reduces SSR by 1%.

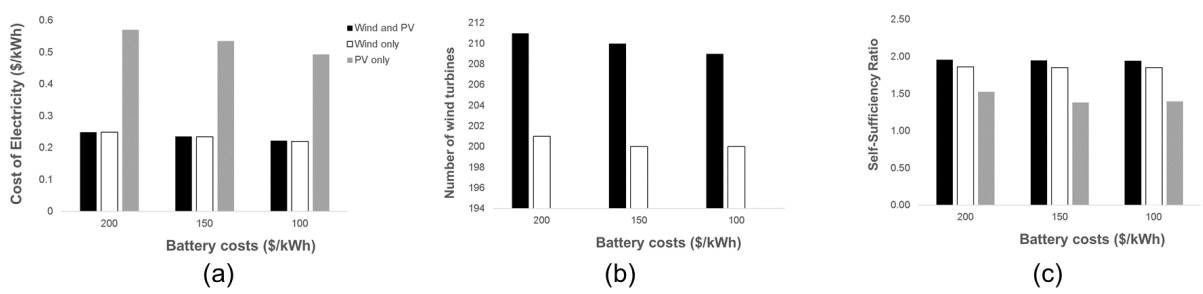

**Figure 8.** Effect of reducing battery capital costs on (a) Cost of Electricity, (b) Number of wind turbines, (c) Self-Sufficiency Ratio.

As can be seen in Fig. 9(a), reducing the wind turbine capital costs from 4,000 $/kW to 1,000 $/kW leads to a significant reduction in COE for the *Wind only* and *Wind and PV* combinations. Halving the wind turbine capital costs reduces COE by

30%, and reducing the wind turbine capital costs to the same value as megawatt-size wind turbines (about 1,000 $/kW) leads to a very reasonable COE of 0.14 $/kWh for the *Wind and PV* combination. Figure 9(b) shows that reducing the wind turbine capital costs leads to an increase in the number of wind turbines chosen for the optimal solution, as expected. Halving the wind turbine capital costs increases the number of wind turbines by 16% for the *Wind and PV* combination. However, Fig. 9(c) shows that reducing the wind turbine capital costs leads to an increase in the SSR. Halving the wind turbine capital costs

increases the SSR by 16% for the *Wind and PV* combination. This is because the larger number of wind turbines lead to a more fluctuating electricity production, making it more difficult to match supply and demand.

As can be seen in Fig. 10(a), reducing the PV capital costs from 3,000 $/kW to 2,000 $/kW leads to a small reduction in COE for the *PV only* and *Wind and PV* combinations. Reducing the PV capital costs by 50% reduces COE by 8% for the *Wind and PV* combination. Figure 10(b) shows that reducing the PV capital costs leads to a significant reduction in the number of

wind turbines chosen for the optimal solution, as expected, as more PV panels are chosen for the optimal solution. Reducing the PV capital costs by 50% decreases the number of wind turbines by 39% for the *Wind and PV* combination. Additionally, it can be seen in Fig. 10(c) that reducing the PV capital costs leads to a significant decrease in the SSR for the *Wind and PV* combination. Reducing the PV capital costs by 50% decreases the SSR by 19% for the *Wind and PV* combination. This is



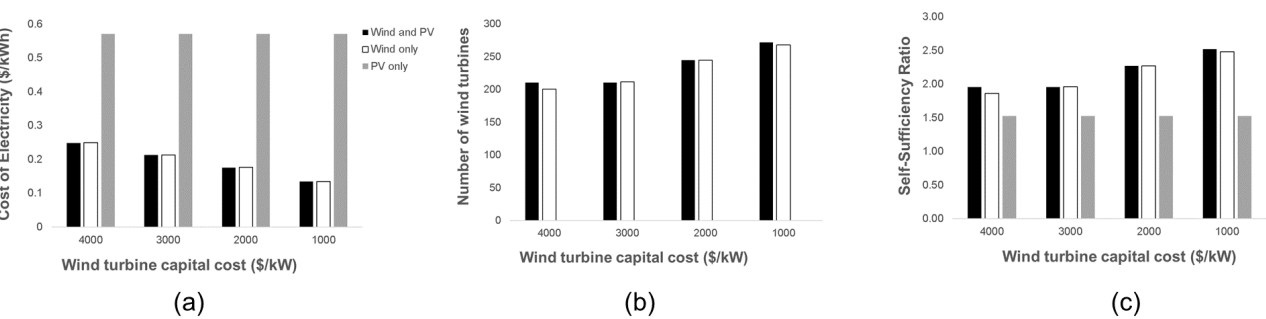

**Figure 9.** Effect of reducing wind turbine capital costs on (a) Cost of Electricity, (b) Number of wind turbines, (c) Self-Sufficiency Ratio.

because the larger number of PV panels and more battery storage combined with fewer wind turbines leads to a more constant electricity production, making it easier to match supply and demand.

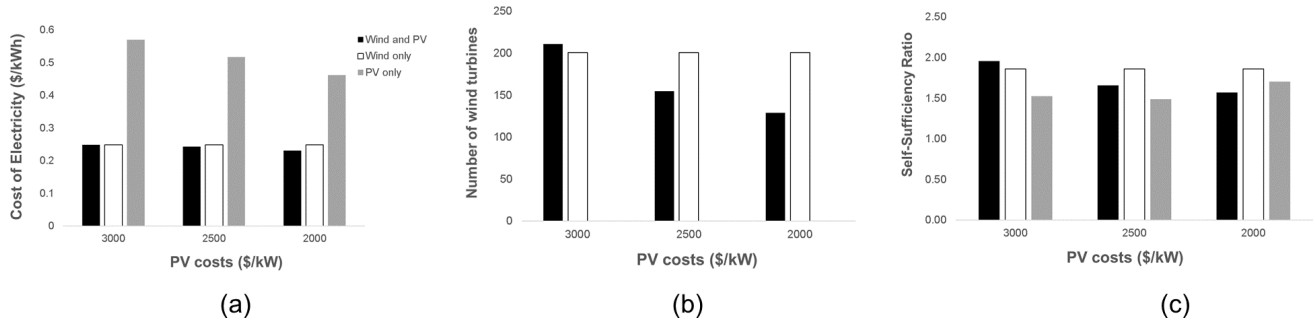

**Figure 10.** Effect of reducing PV capital costs on (a) Cost of Electricity, (b) Number of wind turbines, (c) Self-Sufficiency Ratio.

The results of these sensitivity studies are summarised in Fig. 11. This shows the relative effects of changing the wind turbine, PV and battery costs on the relative COE, SSR and number of wind turbines (WTGs). In summary, reducing the wind turbine costs has a very large relative impact on the COE reduction, whereas reducing the PV costs has a large impact on the number of wind turbines.

### 3.4 Limiting factors

The results shown above were all for load profile number 7 (the highest load, 16,000 kWh/day), but the results were not found to be strongly dependent on the size of the demand. As mentioned in the section above, one limiting factor relating to the size of the demand that can be covered is the **land area** required for the wind turbines and PV panels. The available space is dependent on the project; however, an approximate maximum feasible area was defined in this work to be **100,000 m²** by examining the



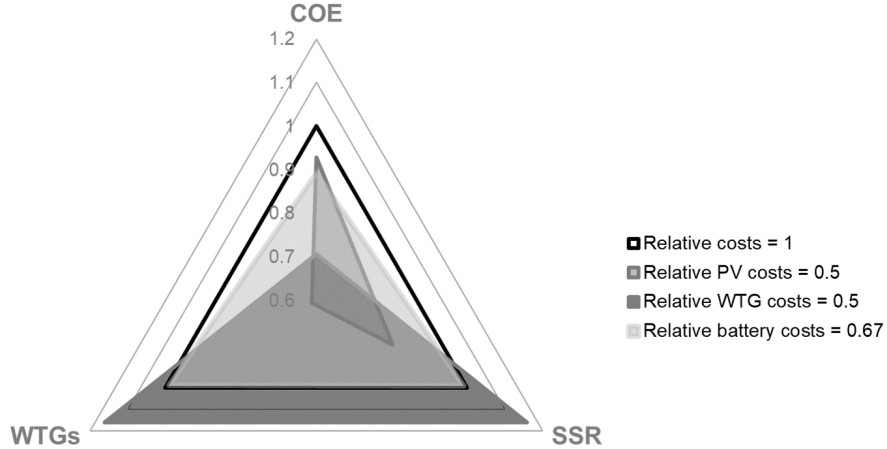

**Figure 11.** Summary of the results of the sensitivity study.

map close to the chosen sites as shown in Fig. 12 for two examples, Wattwil and Les Pleiades. The required space for each solution resulting from the optimisations carried out for this study was calculated based on the following assumptions:

- The wind turbines are separated by ten rotor diameters in one direction and three rotor diameters in the perpendicular direction.

- The PV panels produce 180 W/m$^2$.

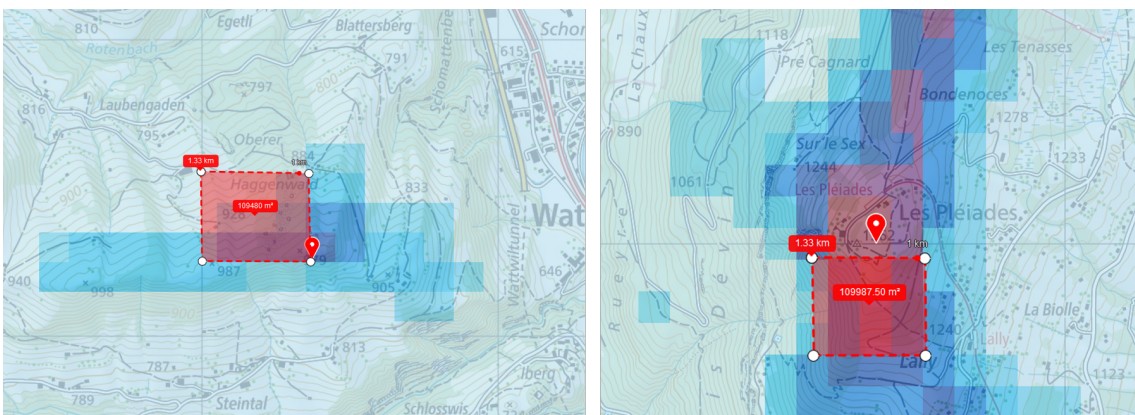

**Figure 12.** Example of maximum available area at two sites, Wattwil (left) and Les Pleiades (right) (*www.wind-data.ch*).

Each solution was then examined in terms of the approximate required area, as shown in Fig. 13(a) for three sites, Grabs (smallest area), Krinau (medium area) and Les Pleiades (largest area). The results for all the other sites were within these




maximum and minimum ranges, but are not shown for clarity. Areas above the maximum defined value are marked by the grey rectangle. The load profiles have been converted to number of four-room houses by assuming that a four-room house demands 2,500 kWh/year (the average value in Canton St. Gallen in Switzerland, Elcom (2019)). It can be seen that load profile 6 and 7 (10,000 and 16,000 kWh/day) are not feasible from the point of view of the maximum area.

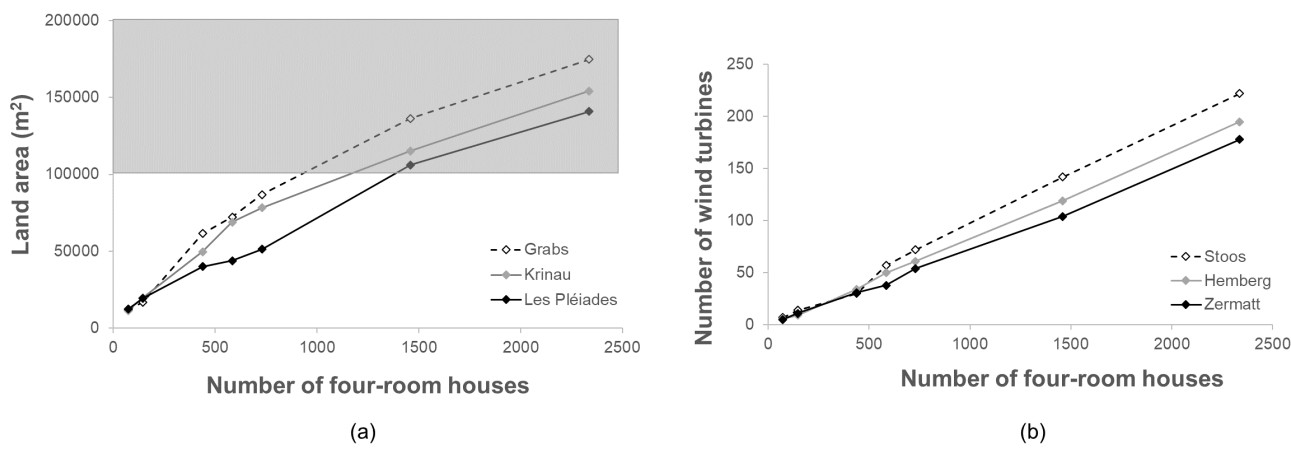

**Figure 13.** (a) Land area and (b) Number of wind turbines vs. number of four-room houses supplied for three sites.

Another limiting factor is the **number of wind turbines** required for the system. The solutions shown above for a load of 16,000 kWh/day require between 180 and 220 medium-sized wind turbines, depending on the site! As well as the amount of area required for this solution, it is questionable whether a community would prefer to install 200 medium-sized wind turbines than one very large wind turbine. The number of wind turbines for each load case is shown in Fig. 13(b) for the sites Zermatt (lowest number), Hemberg (medium number) and Stoos (highest number). The results for all the other sites were

within these maximum and minimum ranges, and are not shown for clarity. It can be seen that only the smallest two load cases are realistically feasible, with five and ten wind turbines, respectively. The larger load cases require more than 30 wind turbines, which does not seem feasible from an acceptance point of view. However, specific projects should be handled on a case-to-case basis in further work.

### 3.5    Case study for High Performance Computing centre

In this case study, a High Performance Computing (HPC) centre in Canton Glarus belonging to the company *ungleich GmbH* was considered. The company is planning an extension to their HPC centre and are interested in supplying it with 100% local renewable energy. They are prepared to pay on the order of 0.15-0.20 $/kWh extra in order to achieve this. The HPC centre consists of 30 servers, each with a 1.1 kW peak, and an estimated 440,000 kWh/year of constant demand. A renewable microgrid system was designed for two nearby locations using the assumptions and methods described above. The first location

was in the valley and the second one was on a nearby hill. Figure 14 shows the results for this case study in terms of the extra





costs per kWh and the number of wind turbines. It can be seen that the site on the hill with the combination *Wind and PV* may be feasible, as the extra costs correspond to approximately 0.2 $/kWh. Furthermore, a total of ten wind turbines seems reasonable. Applying the assumptions for the required area as described in the previous section, this results in a total area of 25,000 m$^2$. An examination of the space available on the map at the location of the hill shows that enough space could be

theoretically available for this system. This must be further studied in terms of planning permission and land ownership.

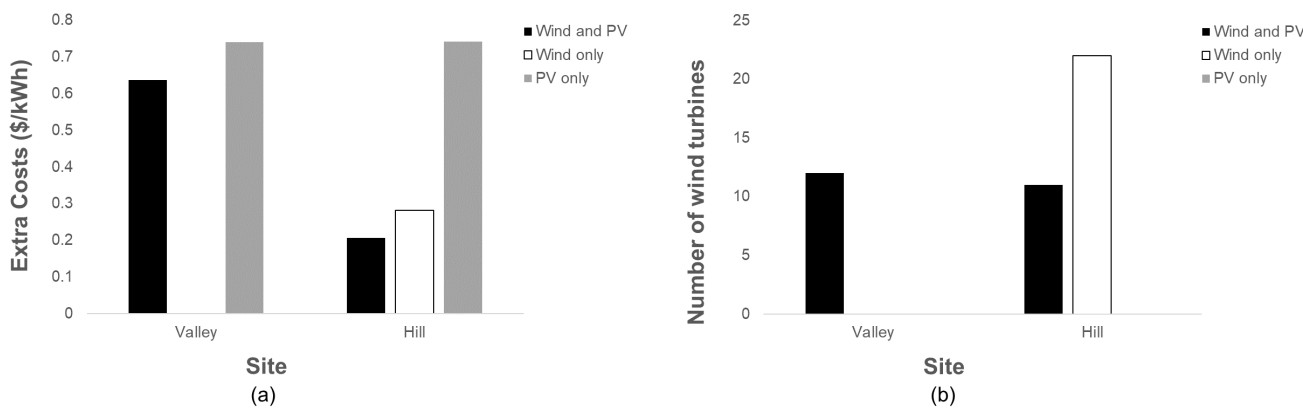

**Figure 14.** (a) Extra costs and (b) Number of wind turbines vs. number of four-room houses supplied for the HPC case study.

## 4   Conclusions

–   In this work, twelve sites in Switzerland were chosen for a 100% renewable energy microgrid feasibility study using medium-sized wind turbines, PV and battery storage.

–   For all of these sites, a combination of wind and PV performed consistently better than wind only and PV only. This is

due to the fact that wind speeds are often higher when the solar radiation is low, and vice versa. The combination of wind and PV ensures a more constant coverage of renewable energy production and therefore is more efficient.

–   Five of the sites were found to be potentially economically viable, if investors would be prepared to make extra investments between 0.05 $/kWh and 0.2 $/kWh or between $5 million and $20 million upfront for green electricity independence.

–   The Self-Sufficiency Ratio was found to range between 1 and 2 for each site, reflecting the extra installed capacity required in order to fully cover every hour of demand in island operation. This could be decreased by connecting to the grid at times of low wind and solar resource and high demand.

–   For the *Wind and PV* combination, halving battery capital costs reduces Costs of Electricity (COE) by 11%, decreases the number of wind turbines by 1% and reduces SSR by 1%.Halving the wind turbine capital costs reduces COE by



30%, increases the number of wind turbines by 16% and increases the SSR by 16%. Reducing the PV capital costs by 50% reduces COE by 8%, decreases the number of wind turbines by 39% and decreases the SSR by 19%.

- The actual implementation of 100% renewable energy microgrids with medium-sized wind turbines was found to be strongly limited by the area required by the wind turbines as well as by the total number of wind turbines that can be realistically implemented.

- A case study was undertaken for an extension to a High Performance Computing centre in Canton Glarus, and a feasible solution was found. Specific projects are being further examined on a case-to-case basis.

*Acknowledgements.* This work was supported by the Energieagentur St. Gallen.



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
