# Peer review of "Feasibility study for 100% renewable energy microgrids in Switzerland"

_Wind Energy Science, 2019_

## Referee Comment (RC1) · Peiyuann Chen (Referee) · 18 Jan 2020

General comments The paper evaluates microgrid sizing problems in Switzerland by using the commercial software HOMER pro. The topic is practical and worth further investigation. However, the analysis lacks depths. The paper resembles a report for pre-study of a microgrid feasibility study, and is suitable for further discussion with industries. For an academic journal, a significant more higher level of analysis and theoretical understanding is needed. This, however, does not imply that the work carried out in the paper is not valuable. The reviewer hopes that the following comments will help to improve the quality of the paper.

Specific comments 2.1 Typically, Levelized cost of electricty is used to measure the

cost of a generation technology. Why would the authors use cost of electricity instead?

2.2. Page 9, line 4, the paper writes 'For an island mode system, it is therefore important that SSR is above one at all times'. I think the sentence does not convey the right message.If SSR<1 for certain hours, it should not be a problem, as the energy storage should cover this temporary power deficiency. But if SSR >1 at all time, isn't the generation system in this island over-dimensioned?

2.3 Page 9, line 11-12, the paper writes 'This is because the combination of wind and PV. allows the set-up to more efficiently cover the production demands due to the complementary nature of the solar and wind resources'.

Can the authors elaborate on this? To what extent is this complementary nature? How to quantify this? Is this level of complementary general in Switzerland, or even in Europe? For a journal, the reviewer would expect a deeper analysis on this, especially this statement serves as the main explanation for the key conclusion of the paper.

2.4 Page 9, Line 12-13, the paper writes 'Furthermore, for nearly all the sites, the Wind only set-up has the highest SSR'. This is because wind power fluctuates more over the year that solar power does'. The annual SSR is only related to the annual energy production, why would this be related to hour-to-hour variation? It is hard to see the cause-effect relation here.

2.5 The area considered is 100,000 m^2, is it reasonable to install 180 - 220 wind turbines? Is noise a problem? Have the authors discussed this analysis with microgrid business developers? I believe this would be useful if possible.

Technical corrections 3.1. Page 9, line 9, 16,000 kWh instead of kW/h.

---

## Referee Comment (RC2) · BJÖRN ANDRESEN (Referee) · 12 Feb 2020

Review of "feasibility Study for 100 % renewable energy microgrid in Switzerland"

Overall I agree to the comments from Peiyuann Chen, the topic is very interesting and needs more focus, some additional points from my side are:

1) You focus mainly on COE, but the main parameter for the decision to invest Is the Levelized cost of energy LCOE. 2) The prices you are assuming for Wind, PV and battery systems are relatively high. Table 1 – the real prizes are around half of the mentioned prices. 3) Table 2 – you use a very simplified load profile for the feasibility study as well as for the house consumption, why not use the well-defined standard load profiles with exist for different consumption types etc. It's seems furthermore also

relatively low to have a electrical average consumption of 2500 kWh / year per house. // In 2015 per capita electricity consumption in Switzerland was 7,033 kWh 4) Self Sufficiency ratio – What have you use as the time scale to evaluate the SSR – yearly , monthly, daily, hourly, minute ? 5) Conclusion – out of this study you will not be able to evaluate if the ,

And in general be a little bit more critical of the results, and it's not "just an upscaling" energy and power are to different parts, it can be that the supply on yearly basis can be provide by Wind, PV and batteries, but you need also more additional power and reserve power – exchange capacities, frequency stability mechanism etc. to provide a stable power system.

Best regards,

Björn

---

## Author Comment (AC1) · 10 Mar 2020

Dear reviewers,

Thanks for taking the time to read the paper and make some useful comments. Here are my answers:

Björn Andresen:  1) It is actually the LCOE that I am calculating.  I have changed the description of this in the final manuscript.  2) The price of wind energy increases with decreasing wind turbine size.  Actually the price of 4,400$/kWh, which comes from a real wind turbine, is very low compared to other small wind turbines (https://www.nrel.gov/analysis/tech-cost-dg.html).  I have checked and corrected the PV and battery costs in the final manuscript. 3) I have checked and adjusted this in

the final manuscript. 4) I have included a more detailed analysis and discussion of the SSR vs. time as well as comparisons of wind and PV production depending on the load vs. time in the final version.

Peiyuann Chen: 2.1 It is actually the LCOE that I am calculating. I have changed the description of this in the final manuscript. 2.2 I have included a more in-depth analysis of SSR in the final version, including an investigation and comparison of the different production vs. time and dependent on the load profiles. 2.3 I have included a more in-depth analysis of this complementary effect in the final version. This includes comparisons of production from PV and wind at different times and depending on the loads. 2.4 This should be answered now too in the final version. 2.5 The answer to this question is already partly given in Figure 13. This shows which solutions work based on this land area. However, I have added a discussion regarding other factors such as noise, permitting, acceptance, etc. to the final version of the manuscript. 3.1 Corrected in the final version.

In general, we have added a significant amount of more discussion and analysis to this paper, especially regarding the SSR and the complementary nature of wind and PV vs. time and depending on demand, and hope that it will now meet the requirements of this journal.

Best regards,

Sarah Barber